

# Gene polymorphisms in *ULK1* and *PIK3CA* are associated with the risk of microscopic polyangiitis in the Guangxi Zhuang Autonomous Region in China

Yan Zhu[1,2], Jinlan Rao[1], Jingsi Wei[1], Liu Liu[1], Shanshan Huang[1], Jingjing Lan[1], Chao Xue[1] and Wei Li[1]

[1] The Second Affiliated Hospital of Guangxi Medical University, Department of Nephrology, Nanning, Guangxi, China
[2] The First Affiliated Hospital, Department of Nephrology, Hengyang Medical School, University of South China, Hengyang, Hunan, China

Corresponding authors
Chao Xue, xccqh@126.com
Wei Li, liwei030514@126.com

## ABSTRACT

**Background**. Microscopic polyangiitis (MPA) is a systemic autoimmune disease characterized by inflammation of small- and medium-sized blood vessels. Autophagy-related protein polymorphisms are involved in autoimmune disease. The aim of this study was to evaluate the effects of single-nucleotide polymorphisms (SNPs) in the *ULK1* and *PIK3CA* genes on the risk of MPA.

**Method**. A total of 208 patients with MPA and 211 controls in the Guangxi Zhuang Autonomous Region were recruited and analyzed. The SNPs selected were detected by polymerase chain reaction and high-throughput sequencing. The differences in allele and genotype frequency, various genetic models, and stratification analyses were evaluated, haplotype evaluation was performed after linkage disequilibrium analysis, and the interaction between gene alleles was analyzed.

**Results**. A statistically significant difference was detected in the genotypic distribution of two SNPs between the two groups: *ULK1* rs4964879 ($p = 0.019$) and *PIK3CA* rs1607237 ($p = 0.002$). The results of the genetic models revealed that *ULK1* rs4964879 and rs9481 were statistically significantly associated with an increased risk of MPA, whereas *PIK3CA* rs1607237 was associated with a reduced risk. The association between SNPs and MPA risk was affected by age, sex, and ethnicity. The *ULK1* haplotype (G-T-A-C-G-A) and *PIK3CA* haplotype (T-G) were associated with a reduced risk of MPA, while the *PIK3CA* haplotype (C-G) was associated with an increased risk.

**Conclusion**. In this study, polymorphisms in the autophagy-related genes *ULK1* and *PIK3CA* and their association with MPA were examined. The results showed that the polymorphisms in *ULK1* (rs4964879 and rs9481) and *PIK3CA* (rs1607237) were significantly associated with MPA risk in the Guangxi population. However, the molecular mechanisms are still unclear; basic science research and studies with larger samples are needed to confirm our conclusions and explore the underlying mechanisms.

## INTRODUCTION

Antineutrophil cytoplasmic antibody (ANCA)-associated vasculitis (AAV) is a group of autoimmune diseases characterized by the inflammation of small- and medium-sized blood vessels. AAV is divided into granulomatosis with polyangiitis (GPA), microscopic polyangiitis (MPA), and eosinophilic GPA (EGPA). Each clinical phenotype is associated with the presence of circulating ANCAs, mainly proteinase-3 (PR3) and myeloperoxidase (MPO) (*Ramponi et al., 2021*). AAV is more common in males than females, and its incidence rate increases with age, especially in the range of 60–70 years (*Geetha & Jefferson, 2020*). Additionally, a notable geographic preponderance of MPA rather than GPA and EGPA is observed in China (*Chang et al., 2019*).

The precise mechanism of AAV remains unestablished, but a genome-wide association study (GWAS) identified the genetic component in the development of this autoimmune disease. A GWAS conducted in a North American cohort demonstrated that GPA is significantly associated with single-nucleotide polymorphisms (SNPs) in the HLA region encoding major histocompatibility complex (MHC) Class II (*Xie et al., 2013*). A European study showed that the genetic association with AAV is antigen specificity. PR3-ANCA is associated with HLA-DP, which encodes SERPINA1 and PRTN3, while MPO-ANCA is associated with HLA-DQ (*Lyons et al., 2012*). Consistent with previous studies, a new large GWAS revealed that MHC and non-MHC gene variates are related to GPA/MPA susceptibility, and changing the expression of genes and proteins is associated with the immune response (*Merkel et al., 2017*).

Autophagy is an essential metabolic process in eukaryotic cells, and autophagy-related proteins are involved in various pathologies, including disorders of immune regulation, inflammation, and cancer (*Wu & Adamopoulos, 2017*). The PI3K/AKT/mTOR/ULK1 signaling pathway is one of the main regulators of autophagy. Uncoordinated 51-like kinase 1 (ULK1) is a serine/threonine kinase that plays a key role in the formation of the ULK1 complex. The human ULK1 complex can induce the initiation of the autophagy pathway and maturation of autophagosomes (*Lin & Hurley, 2016*). PIK3CA encodes the p110 $\alpha$ catalytic subunit of phosphatidylinositol 3-kinase (PI3K), which can inactivate mTOR through the PIK3CA/AKT signaling pathway and lead to autophagy (*Qu et al., 2016*). PIK3CA and ULK1 are the core components of this signaling pathway, and their mutations may alter the autophagy response and cause a change in the incidence of disease risk (*Morgan et al., 2012*; *Qu et al., 2016*; *Zhang & Zhou, 2019*; *Zhang et al., 2017*).

Numerous studies have revealed that autoimmune diseases may share susceptibility genes. *ULK1* has been shown to be associated with ankylosing spondylitis (*Zhang et al., 2017*) and Crohn's disease (*Morgan et al., 2012*). The cooccurrence of systemic lupus erythematosus (SLE) and AAV reported in cases suggests that these two diseases may have shared genetic factors, especially in MPO-ANCA-positive AAV (*Hervier et al., 2012*). Emerging evidence shows that autophagy-related gene polymorphisms, such as mTOR (*Saravani et al., 2020*) and ATG5 (*López et al., 2013*), participate in SLE. However, as an autoimmune disease, the role of autophagy-related gene mutations in AAV has not yet been reported. Considering that MPA is the most common clinical subtype of AAV in China, we

focused on whether gene polymorphisms of *ULK1* and *PIK3CA* play a role in susceptibility to MPA. In the present study, SNP loci with a minor allele frequency (MAF) ≥5% in the functional region of the *ULK1* and *PIK3CA* genes were selected, and the association between these two gene polymorphisms and susceptibility to MPA was explored in a group of patients with MPA and a healthy control group from the Guangxi Zhuang Autonomous Region in China.

## MATERIALS & METHODS

### Study population

A total of 208 eligible patients with MPA were recruited from September 2009 to April 2020 in the Department of Nephrology of the Second Affiliated Hospital of Guangxi Medical University (formerly Western Hospital of the First Affiliated Hospital of Guangxi Medical University). The inclusion criteria were as follows: (i) all cases were classified and evaluated as MPA according to the 2012 Revised International Chapel Hill Consensus Conference Nomenclature of Vasculitis (*Jennette et al., 2013*), (ii) age ≥18 years, and (iii) all patients were born in the Guangxi Zhuang Autonomous Region and had no blood relationship. Patients with secondary vasculitis, other autoimmune diseases, chronic disease and malignant tumors were excluded. A total of 211 healthy volunteers matching the MPA group with respect to age and sex were enrolled as the control group.

The basic clinical information of the patients with MPA and the healthy controls is presented in Table 1. The age range at presentation was 18–82 years, with a mean age of 54.6 ± 14.9 years, of which 114 cases were <60 years, and 62.5% were female. The MPA group had 131 Han and 75 Zhuang nationality populations. The mean BVAS at diagnosis was 16.8 ± 4.43. In this study, 36 biopsy specimens (40.4%) were classified as focal, 9 (10.1%) as crescentic, 20 (22.5%) as mixed, and 24 (27%) as sclerotic. Tubulointerstitial injury was graded as follows: 37 (41.6%) had a score of 1, 41 (46.1%) had a score of 2, and 11 (12.4%) had a score of 3. The control group (mean age 51.2 ± 12.6 years) consisted of 128 females and 155 Han nationality populations. This study was approved by the Ethics Committee of the Second Affiliated Hospital of Guangxi Medical University (No. 2018 KY-0100) and followed the principles of the Helsinki Declaration. Written informed consent was obtained from all participants.

### DNA isolation

Blood (5 ml) was collected from the ulnar vein of each participant. Total genomic DNA was extracted from peripheral blood samples using a blood DNA extraction kit (Tiangen, Beijing, China) according to the manufacturer's instructions, and the quality was checked by a Nanodrop 2000 spectrophotometer (Thermo Scientific). Samples with an A260/A280 ratio of 1.7–1.9 were included in the study, and the isolated DNA was stored at −80 °C for further studies.

### Tag SNP selection

Six SNPs of the *ULK1* gene (rs10902469, rs12303764, rs4964879, rs7300908, rs7138581 and rs9481) and two SNPs of the *PIK3CA* gene (rs1607237 and rs9838117) were selected

Table 1 Demographic characteristic of the study participants.

| Characteristic | MPA group (n = 208) | Control group (n = 211) |
|---|---|---|
| Age (years) | 54.6 ± 14.9 | 51.2 ± 12.6 |
| <60 | 114 (54.8) | 160 (75.8) |
| ≥60 | 94 (45.2) | 51 (24.2) |
| Sex (M/F) | 78/130 | 83/128 |
| Ethnicity (Han/Zhuang) | 131/75 | 155/56 |
| BVAS (mean ± SD) | 16.8 ± 4.43 | – |
| Renal pathologic classification (Renal biopsy, n = 89) | | |
| Focal | 36 (40.4%) | |
| Crescentic | 9 (10.1%) | |
| Mixed | 20 (22.5%) | |
| Sclerotic | 24 (27.0%) | |
| Renal tubulointerstitial injury (Renal biopsy, n = 89) | | |
| Score 1 | 37 (41.6%) | |
| Score 2 | 41 (46.1%) | |
| Score 3 | 11 (12.4%) | |

from genotype data of Chinese people in the 1000 Genomes (http://grch37.ensembl.org/). The selection criteria included the following: (1) sites located in the functional region, (2) previously reported associations with autoimmune or inflammatory diseases, (3) select tag SNPs as determined using HaploReg, and (4) MAF ≥0.05.

## SNP genotyping assay

SNPs of the *ULK1* and *PIK3CA* genes were detected by polymerase chain reaction (PCR) and high-throughput sequencing (Sangon Biotech, Shanghai, China). The PCR amplification conditions were settled by the two-step method. HiSeq XTen sequencers (Illumina, San Diego, CA, USA) were used to perform paired-end sequencing of the library, and the data were analyzed using Samtools 0.1.18 software. Approximately 10% of the randomly selected samples were sequenced by Sangon Biotechnology Company (Shanghai, China) to verify the accuracy of genotyping, and the reproducibility rate of all SNP genotyping was 100%.

## Statistical analysis

The genotypic and allelic frequencies in the MPA group and the control group were evaluated by the chi-square test or Fisher's exact test. Hardy–Weinberg equilibrium (HWE) in the control participants was tested using the chi-square test for each SNP. Genetic models and stratification analyses with odds ratios (ORs) and 95% confidence intervals (CIs) were analyzed to estimate the relationship between genetic variation and the risk of MPA through online SNPstats software (https://www.snpstats.net/start.htm) adjusted by age and sex. Pairwise linkage disequilibrium (LD) and haplotype blocks as measured by D' were evaluated by online software (SHEsis) (*Shi & He, 2005*). The interactions between

SNPs of the *ULK1* gene and *PIK3CA* gene were evaluated using generalized multifactor dimensionality reduction (GMDR). SPSS Statistics version 23.0 (IBM, Armonk, NY, USA) was used to analyze the data, and $p < 0.05$ was considered statistically significant.

## RESULTS

### Association of gene polymorphisms with MPA susceptibility

The genotyping results for quality control ranged from 97.18% to 99.76%. Detailed information on all SNPs is provided in Table 2 (SNP IDs, locations and allele frequencies). In the selected SNPs, all ANPs had a MAF of >5%, and the genotype distribution in the control group was in HWE ($p > 0.05$). According to the single-SNP analyses, the allele frequencies of *PIK3CA* rs1607237 (C>T) were significantly different between the MPA group and the control group ($p = 0.011$).

The association between the SNPs and the risk of MPA was identified by genetic models (codominant, dominant, recessive, and overdominant) and genotype frequencies (Table 3). The results adjusted by age and sex showed that rs4964879 in the *ULK1* gene significantly increased the risk of MPA with the GA genotype in the codominant model (GA versus AA, OR = 1.76, 95% CI [1.15–2.70], and $p = 0.03$), the dominant model (GA/GG versus AA, OR = 1.60, 95% CI [1.07–2.40], and $p = 0.022$) and overdominant model (GA versus AA/GG, OR = 1.68, 95% CI [1.13–2.49], and $p = 0.0096$). The risk of MPA in the *ULK1* gene rs9481 was 1.77 times that in healthy controls in the recessive model (GG versus AA/AG, 95% CI [1.06–2.94], and $p = 0.027$). The mutations of rs1607237 in the *PIK3CA* gene had a lower incidence of MPA with the CT genotype in the codominant model (CT versus CC, OR = 0.47, 95% CI [0.30–0.73], and $p = 0.0039$), the dominant model (CT/TT versus CC, OR = 0.55, 95% CI [0.37–0.82], and $p = 0.0031$), and the overdominant model (CT versus CC/TT, OR = 0.49, 95% CI [0.32–0.76], and $p = 0.0013$). No significant difference was observed for the other gene loci between the cases and controls ($p > 0.05$).

### Linkage disequilibrium analysis

Figure 1 shows the pattern of pairwise LD with respect to the analyzed SNPs of two genes in the current study. The LD plot indicated that the *ULK1* rs10902469, rs12303764, rs4964879, rs7300908, rs7138581, and rs9481 loci formed six haplotypes (Table 4). Haplotype G-T-A-C-G-A was the most commonly observed haplotype in the cases (49.2%) and in the healthy controls (56.1%) and was associated with a reduced risk of MPA (OR = 0.749, 95% CI [0.563–0.997], $p = 0.047$). Other haplotypes did not exhibit an association with MPA. The SNP loci of the *PIK3CA* genes rs1607237 and rs9838117 also formed three haplotypes (Table 4). The results showed that the C-G haplotype was the most commonly observed haplotype in the cases (73.3%) and in the healthy controls (66.7%) and was associated with an increased risk of MPA (OR = 1.427, 95% CI [1.050–1.939], and $p = 0.023$). The T-G haplotype was significantly associated with a reduced risk of MPA (OR = 0.520, 95% CI [0.339–0.799], and $p = 0.0025$).

### Stratification analysis based on age, sex and ethnicity

The analysis results showed that age, sex, and ethnicity significantly affected the association between *ULK1* and *PIK3CA* SNPs and MPA risk. The mutations of *ULK1* rs4964879 (with

**Table 2  Basic information about SNPs in *ULK1* and *PIK3CA* and their association with the risk of MPA.**

| Gene | SNP ID | Location | Alleles | MAF | | $p$ for allele frequencies | $p$ for genotypes |
|------|--------|----------|---------|------|------|------|------|
| | | | | Case | Control | | |
| ULK1 | rs10902469 | 12:132378133 | G>C | 0.079 | 0.069 | 0.571 | 0.832 |
| | rs12303764 | 12:132399065 | T>G | 0.200 | 0.185 | 0.566 | 0.678 |
| | rs4964879 | 12:132400309 | A>G | 0.406 | 0.356 | 0.143 | *0.019* |
| | rs7300908 | 12:132405421 | C>T | 0.099 | 0.090 | 0.673 | 0.897 |
| | rs7138581 | 12:132406666 | G>C | 0.174 | 0.167 | 0.781 | 0.853 |
| | rs9481 | 12:132407089 | A>G | 0.442 | 0.378 | 0.060 | 0.070 |
| PIK3CA | rs1607237 | 3:178950297 | C>T | 0.240 | 0.325 | *0.011* | *0.002* |
| | rs9838117 | 3:178952507 | G>T | 0.168 | 0.165 | 0.901 | 0.765 |

**Table 3  The genotype frequencies of the studied *ULK1* and *PIK3CA* gene SNPs in the cases and the healthy controls.**

| SNP ID | Model | Genotype | Control n (%) | Case n (%) | OR (95% CI) | $p$ |
|--------|-------|----------|---------------|------------|-------------|-----|
| ULK1 rs4964879 | Codominant | AA | 92 (44) | 67 (32.4) | 1.00 | *0.03* |
| | | GA | 85 (40.7) | 112 (54.1) | **1.76 (1.15–2.70)** | |
| | | GG | 32 (15.3) | 28 (13.5) | 1.19 (0.65–2.16) | |
| | Dominant | AA | 92 (44) | 68 (31.9) | 1.00 | *0.022* |
| | | GA+GG | 117 (56) | 145 (68.1) | **1.60 (1.07–2.40)** | |
| | Recessive | AA+GA | 177 (84.7) | 184 (86.4) | 1.00 | 0.61 |
| | | GG | 32 (15.3) | 29 (13.6) | 0.87 (0.50–1.51) | |
| | Overdominant | AA+GG | 124 (59.3) | 97 (45.5) | 1.00 | *0.0096* |
| | | GA | 85 (40.7) | 116 (54.5) | **1.68 (1.13–2.49)** | |
| ULK1 rs9481 | Codominant | AA | 81 (38.8) | 73 (34.3) | 1.00 | 0.086 |
| | | AG | 98 (46.9) | 90 (42.2) | 1.00 (0.65–1.54) | |
| | | GG | 30 (14.3) | 50 (23.5) | 1.76 (1.01–3.09) | |
| | Dominant | AA | 81 (38.8) | 73 (34.3) | 1.00 | 0.36 |
| | | AG/GG | 128 (61.2) | 140 (65.7) | 1.18 (0.79–1.76) | |
| | Recessive | AA/AG | 179 (85.7) | 163 (76.5) | 1.00 | *0.027* |
| | | GG | 30 (14.3) | 50 (23.5) | **1.77 (1.06–2.94)** | |
| | Overdominant | AA/GG | 111 (53.1) | 123 (57.8) | 1.00 | 0.34 |
| | | AG | 98 (46.9) | 90 (42.2) | 0.83 (0.56–1.22) | |
| PIK3CA rs1607237 | Codominant | CC | 101 (48.6) | 138 (64.3) | 1.00 | *0.0039* |
| | | CT | 79 (38) | 45 (22.6) | **0.47 (0.30–0.73)** | |
| | | TT | 28 (13.5) | 26 (13.1) | 0.78 (0.43–1.42) | |
| | Dominant | CC | 101 (48.6) | 128 (64.3) | 1.00 | *0.0031* |
| | | CT/TT | 107 (51.4) | 71 (35.7) | **0.55 (0.37–0.82)** | |
| | Recessive | CC/CT | 180 (86.5) | 173 (86.9) | 1.00 | 0.97 |
| | | TT | 28 (13.5) | 26 (13.1) | 1.01 (0.57–1.81) | |
| | Overdominant | CC/TT | 129 (62) | 154 (77.4) | 1.00 | *0.0013* |
| | | CT | 79 (38) | 45 (22.6) | **0.49 (0.32–0.76)** | |

GA genotype in the overdominant model, OR = 1.65, 95% CI [1.01–2.69], and $p = 0.046$) and rs9481 (with GG genotype in the recessive model, OR = 1.88, 95% CI [1.01–3.51], and $p = 0.047$) were associated with a higher incidence of MPA in the population aged < 60

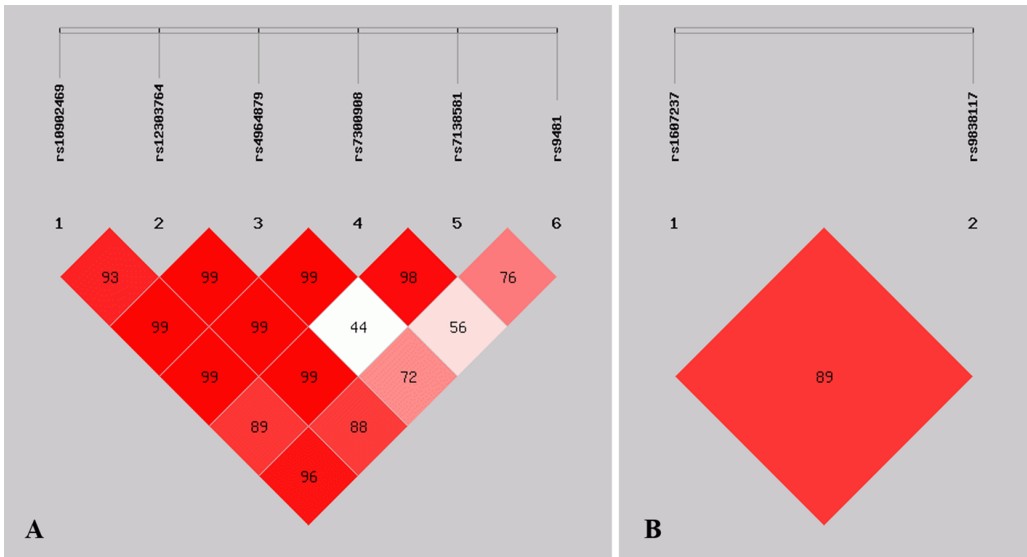

**Figure 1 Graphical representation of the SNP locations and LD structure.** (A) LD plots containing 6 SNPs from *ULK1*; (B) LD plots containing 2 SNPs from *PIK3CA*.

**Table 4 The correlation between the haplotypes of *ULK1* and *PIK3CA* gene SNPs and the MPA susceptibility.**

| Gene | Haplotype | Case (n = 208) | Control (n = 211) | OR (95% CI) | p |
|------|-----------|----------------|-------------------|-------------|---|
| *ULK1* | C-T-A-C-C-G | 27.6 (6.7%) | 27.1 (6.5%) | 1.039 (0.599~1.799) | 0.893 |
| | G-G-G-G-G | 80.8 (19.5%) | 67.0 (16%) | 1.285 (0.897~1.841) | 0.171 |
| | G-T-A-C-G-A | 203.7 (49.2%) | 234.5 (56.1%) | **0.749 (0.563~0.997)** | **0.047** |
| | G-T-G-C-G-A | 13.9 (3.4%) | 10.6 (2.5%) | 1.343 (0.596~3.024) | 0.475 |
| | G-T-G-C-G-G | 26.8 (6.5%) | 20.7 (4.9%) | 1.340 (0.741~2.422) | 0.331 |
| | G-T-G-T-C-G | 30.8 (7.4%) | 29.9 (7.1%) | 1.054 (0.624~1.780) | 0.844 |
| *PIK3CA* | C-G | 291.5 (73.3%) | 278.7 (66.7%) | **1.427 (1.050~1.939)** | **0.023** |
| | T-G | 36.5 (8.8%) | 68.3 (16.3%) | **0.520 (0.339~0.799)** | **0.0025** |
| | T-T | 62.5 (15.7%) | 66.7 (16.0%) | 0.993 (0.681~1.446] | 0.9688 |

years; *PIK3CA* rs1607237 was associated with a decreased MPA risk under the codominant model (CT versus CC, OR = 0.23, 95% CI [0.10–0.53], and $p < 0.001$), the dominant model (CT/TT versus CC, OR = 0.41, 95% CI [0.20–0.85], and $p = 0.016$), and the overdominant model (CT versus CC/TT, OR = 0.21, 95% CI [0.09–0.49], and $p < 0.001$) in the population $\geq 60$ years (Table 5). The results were adjusted by sex.

The results also showed that sex significantly affected the association between SNPs and MPA risk (Table 6). *ULK1* rs4964879 in females under the dominant model (GA/GG versus AA, OR = 1.69, 95% CI [1.02–2.82], and $p = 0.042$) and the overdominant model (GA versus AA/GG, OR = 1.74, 95% CI [1.05–2.88], and $p = 0.031$), which could increase MPA risk. Marginal evidence revealed that rs9481 in females under the regressive model

Zhu et al. (2021), *PeerJ*, DOI 10.7717/peerj.12377

**Table 5  Distribution of *ULK1* and *PIK3CA* polymorphisms in population of different ages and its association with the risk of MPA.**

| SNP ID | Model | Geno type | Age <60 years | | | P value | Age ≥ 60 years | | | p value |
|---|---|---|---|---|---|---|---|---|---|---|
| | | | Control | Case | OR (95% CI) | | Control | Case | OR (95% CI) | |
| | | AA | 72 (45%) | 39 (34.2%) | 1.00 | | 20 (40.8%) | 28 (30.1%) | 1.00 | |
| | Codominant | GA | 62 (38.8%) | 58 (50.9%) | 1.74 (1.02–2.96) | 0.12 | 23 (46.9%) | 54 (58.1%) | 1.66 (0.78–3.53) | 0.42 |
| | | GG | 26 (16.2%) | 17 (14.9%) | 1.21 (0.59–2.49) | | 6 (12.2%) | 11 (11.8%) | 1.37 (0.43–4.35) | |
| | | AA | 72 (45%) | 39 (34.2%) | 1.00 | | 20 (40.8%) | 28 (20.1%) | 1.00 | |
| *ULK1* rs4964879 | Dominant | GA/GG | 88 (55%) | 75 (65.8%) | 1.58 (0.96–2.60) | 0.073 | 29 (59.2%) | 65 (69.9%) | 1.60 (0.77–3.30) | 0.21 |
| | | AA/GA | 134 (83.8%) | 97 (85.1%) | 1.00 | | 43 (87.8%) | 82 (88.2%) | 1.00 | |
| | Recessive | GG | 26 (16.2%) | 17 (14.9%) | 0.90 (0.47–1.76) | 0.77 | 6 (12.2%) | 11 (11.8%) | 1.01 (0.35–2.95) | 0.98 |
| | | AA/GG | 98 (61.2%) | 56 (49.1%) | 1.00 | | 26 (53.1%) | 39 (41.9%) | 1.00 | |
| | Overdominant | GA | 62 (38.8%) | 58 (50.9%) | ***1.65 (1.01–2.69)*** | ***0.046*** | 23 (46.9%) | 54 (58.1%) | 1.53 (0.76–3.08) | 0.23 |
| | | AA | 63 (39.4%) | 40 (35.1%) | 1.00 | | 18 (36.7%) | 32 (34.4%) | 1.00 | |
| | Codominant | AG | 74 (46.2%) | 47 (41.2%) | 0.99 (0.57–1.70) | 0.14 | 24 (49%) | 40 (43%) | 0.92 (0.42–1.99) | 0.44 |
| | | GG | 23 (14.4%) | 27 (23.7%) | 1.87(0.94–3.71) | | 7 (14.3%) | 21 (22.6%) | 1.73 (0.61–4.88) | |
| | | AA | 63 (39.4%) | 40 (35.1%) | 1.00 | | 18 (36.7%) | 32 (34.4%) | 1.00 | |
| *ULK1* rs9481 | Dominant | AG/GG | 97 (60.6%) | 74 (64.9%) | 1.20 (0.73–1.98) | 0.47 | 31 (63.3%) | 61 (65.6%) | 1.10 (0.53–2.27) | 0.8 |
| | | AA/AG | 137 (85.6%) | 87 (76.3%) | 1.00 | | 42 (85.7%) | 72 (77.4%) | 1.00 | |
| | Recessive | GG | 23 (14.4%) | 27 (23.7%) | ***1.88 (1.01–3.51)*** | ***0.047*** | 7 (14.3%) | 21 (22.6%) | 1.81 (0.71–4.65) | 0.2 |
| | | AA/GG | 86 (53.8%) | 67 (58.8%) | 1.00 | | 25 (51%) | 53 (57%) | 1.00 | |
| | Overdominant | AG | 74 (46.2%) | 47 (41.2%) | 0.81 (0.49–1.32) | 0.39 | 24 (49%) | 40 (43%) | 0.77 (0.38–1.54) | 0.45 |
| *PIK3CA* rs1607237 | | CC | 76 (47.8%) | 63 (58.3%) | 1.00 | | 25 (51%) | 65 (71.4%) | 1.00 | |
| | Codominant | CT | 58 (36.5%) | 32 (29.6%) | 0.67 (0.39–1.15) | 0.23 | 21 (42.9%) | 13 (14.3%) | ***0.23 (0.10–0.53)*** | ***6e−04*** |
| | | TT | 25 (15.7%) | 13 (12%) | 0.62 (0.29–1.32) | | 3 (6.1%) | 13 (14.3%) | 1.70 (0.44–6.51) | |
| | | CC | 76 (47.8%) | 63 (58.3%) | 1.00 | | 25 (51%) | 65 (71.4%) | 1.00 | |
| | Dominant | CT/TT | 83 (52.2%) | 45 (41.7%) | 0.65 (0.40–1.07) | 0.09 | 24 (49%) | 26 (28.6%) | ***0.41 (0.20–0.85)*** | ***0.016*** |
| *PIK3CA* rs1607237 | | CC/CT | 134 (84.3%) | 95 (88%) | 1.00 | | 46 (93.9%) | 78 (85.7%) | 1.00 | |
| | Recessive | TT | 25 (15.7%) | 13 (12%) | 0.73 (0.35–1.49) | 0.38 | 3 (6.1%) | 13 (14.3%) | 2.61 (0.70–9.71) | 0.12 |
| | | CC/TT | 101 (63.5%) | 76 (70.4%) | 1.00 | | 28 (57.1%) | 78 (85.7%) | 1.00 | |
| | Overdominant | CT | 58 (36.5%) | 32 (29.6%) | 0.74 (0.44–1.24) | 0.25 | 21 (42.9%) | 13 (14.3%) | ***0.21 (0.09–0.49)*** | ***2e−04*** |

increased MPA risk ($p = 0.05$). *PIK3CA* rs1607237 was associated with a decreased MPA risk in the population, independent of sex. The results were adjusted by age.

In addition, the results showed that the Han population with the *ULK1* rs4964879 mutation had a higher incidence of MPA with the GA genotype under the dominant model (GA/GG versus AA, OR = 1.78, 95% CI [1.09–2.90], and $p = 0.02$, Table 7) and the overdominant model (GA versus AA/GG, OR = 1.70, 95% CI [1.05–2.74], and $p = 0.03$); the Han population with *PIK3CA* rs1607237 could significantly decrease MPA risk with the CT genotype in the codominant model (CT versus CC, OR = 0.42, 95% CI [0.24–0.73], and $p = 0.0068$), the dominant model (CT/TT versus CC, OR = 0.48, 95% CI [0.29–0.78], and $p = 0.0031$) and the overdominant model (CT versus CC/TT, OR = 0.44, 95% CI [0.26–0.76], $p = 0.0026$). The results were adjusted by sex and age.

### Interaction of gene alleles with clinical characteristics

Generalized multifactor dimensionality reduction (GMDR) was used to analyze the interaction between the alleles of the *ULK1* gene (rs10902469, rs12303764, rs4964879, rs7300908, rs7138581, and rs9481) and *PIK3CA* gene (rs1607237 and rs9838117). The interaction showed that rs4964879 and rs1607237 were the best models for MPA prediction (cross-validation consistency: 10/10). The risk of MPA in the "high-risk" combination was 2.27 times that in the "low-risk combination" (Fig. 2), but a margin testing $p$ value was observed ($p = 0.0547$).

## DISCUSSION

In this study, polymorphisms in the autophagy-related genes *ULK1* and *PIK3CA* and their association with MPA were examined. The results showed that the *ULK1* SNPs rs4964879 and rs9481 were risk factors for MPA, and *PIK3CA* rs1607237 was a protective factor for MPA.

Autophagy is a fundamental intracellular biological process of eukaryotic cells that is essential for the activation of innate and adaptive immune responses, including self-antigen presentation, phagocytosis, maintenance of lymphocyte homeostasis, and regulation of cytokine production (*Ye, Zhou & Zhang, 2019*). It is well established that the mammalian target of rapamycin (mTOR)/ULK1 pathway is one of the main regulators of autophagy. Inhibition of mTOR results in dephosphorylation of ULK1 and upregulates autophagy, and it is positively modulated through the PI3K/AKT pathway and negatively modulated by adenosine monophosphate-activated protein kinase (*Mohamed et al., 2021*). Increasing studies have demonstrated that autophagy is involved in the biology of neutrophils, which play a critical role in the acute injury of AAV by releasing proteolytic enzymes via degranulation, producing reactive oxygen species and extruding neutrophil extracellular traps (NETs) (*Al-Hussain et al., 2017*; *Skendros, Mitroulis & Ritis, 2018*). *Sha et al. (2016)* proved that autophagy activity is elevated in neutrophils treated with ANCAs, and the NET formation rate increases or decreases in neutrophils pretreated with an autophagy inducer or inhibitor, respectively. *Tang et al. (2015)* also demonstrated that NET formation is associated with autophagy-related signaling in human neutrophils with AAV.

Zhu et al. (2021), *PeerJ*, DOI 10.7717/peerj.12377

**Table 6 Distribution of *ULK1* and *PIK3CA* polymorphisms in population of different sex and its association with the risk of MPA.**

| SNP ID | Model | Geno-type | Male | | | *p* value | Female | | | *p* value |
|---|---|---|---|---|---|---|---|---|---|---|
| | | | Control | Case | OR (95% CI) | | Control | Case | OR (95% CI) | |
| | Codominant | AA | 31 (37.8%) | 23 (29.1%) | 1.00 | 0.32 | 61 (48%) | 44 (34.1%) | 1.00 | 0.08 |
| | | GA | 38 (46.3%) | 46 (58.2%) | 1.63 (0.82–3.26) | | 47 (37%) | 67 (51.9%) | 1.86 (1.08–3.20) | |
| | | GG | 13 (15.8%) | 10 (12.7%) | 1.03 (0.38–2.77) | | 19 (15%) | 18 (13.9%) | 1.28 (0.60–2.73) | |
| *ULK1* rs4964879 | Dominant | AA | 31 (37.8%) | 23 (29.1%) | 1.00 | 0.25 | 61 (48%) | 44 (34.1%) | 1.00 | 0.042 |
| | | GA/GG | 51 (62.2%) | 56 (70.9%) | 1.47 (0.76–2.86) | | 66 (52%) | 85 (65.9%) | *1.69 (1.02–2.82)* | |
| | Recessive | AA/GA | 69 (84.2%) | 69 (87.3%) | 1.00 | 0.56 | 108 (85%) | 111 (86%) | 1.00 | 0.84 |
| | | GG | 13 (15.8%) | 10 (12.7%) | 0.76 (0.31–1.87) | | 19 (15%) | 18 (13.9%) | 0.93 (0.46–1.88) | |
| | Overdominant | AA/GG | 44 (53.7%) | 33 (41.8%) | 1.00 | 0.13 | 80 (63%) | 62 (48.1%) | 1.00 | 0.031 |
| | | GA | 38 (46.3%) | 46 (58.2%) | 1.61 (0.86–3.02) | | 47 (37%) | 67 (51.9%) | *1.74 (1.05–2.88)* | |
| | Codominant | AA | 29 (35.4%) | 28 (35.4%) | 1.00 | 0.5 | 52 (40.9%) | 45 (34.9%) | 1.00 | 0.14 |
| | | AG | 43 (52.4%) | 36 (45.6%) | 0.89 (0.45–1.76) | | 55 (43.3%) | 51 (39.5%) | 1.04 (0.60–1.82) | |
| | | GG | 10 (12.2%) | 15 (19%) | 1.49 (0.57–3.90) | | 20 (15.8%) | 33 (25.6%) | 1.90 (0.95–3.78) | |
| *ULK1* rs9481 | Dominant | AA | 29 (35.4%) | 28 (35.4%) | 1.00 | 0.86 | 52 (40.9%) | 45 (34.9%) | 1.00 | 0.36 |
| | | AG/GG | 53 (64.6%) | 51 (64.6%) | 1.00 (0.52–1.92) | | 75 (59.1%) | 84 (65.1%) | 1.27 (0.76–2.12) | |
| | Recessive | AA/AG | 72 (87.8%) | 64 (81%) | 1.00 | 0.25 | 107 (84.2%) | 96 (74.4%) | 1.00 | 0.05 |
| | | GG | 10 (12.2%) | 15 (19%) | 1.60 (0.67–3.84) | | 20 (15.8%) | 33 (25.6%) | *1.86 (0.99–3.47)* | |
| | Overdominant | AA/GG | 39 (47.6%) | 43 (54.4%) | 1.00 | 0.51 | 72 (56.7%) | 78 (60.5%) | 1.00 | 0.48 |
| | | AG | 43 (52.4%) | 36 (45.6%) | 0.79 (0.42–1.47) | | 55 (43.3%) | 51 (39.5%) | 0.83 (0.50–1.38) | |
| *PIK3CA* rs1607237 | Codominant | CC | 39 (47.6%) | 51 (66.2%) | 1.00 | 0.042 | 62 (49.2%) | 78 (63.4%) | 1.00 | 0.048 |
| | | CT | 29 (35.4%) | 15 (19.5%) | *0.40 (0.19–0.84)* | | 50 (39.7%) | 30 (24.4%) | *0.50 (0.28–0.88)* | |
| | | TT | 14 (17.1%) | 11 (14.3%) | 0.60 (0.25–1.47) | | 14 (11.1%) | 15 (12.2%) | 0.93 (0.41–2.11) | |
| | Dominant | CC | 39 (47.6%) | 51 (66.2%) | 1.00 | 0.017 | 62 (49.2%) | 78 (63.4%) | 1.00 | 0.063 |
| | | CT/TT | 43 (52.4%) | 26 (33.8%) | *0.46 (0.24–0.88)* | | 64 (50.8%) | 45 (36.6%) | *0.59 (0.35–0.99)* | |
| *PIK3CA* rs1607237 | Recessive | CC/CT | 68 (82.9%) | 66 (85.7%) | 1.00 | 0.63 | 112 (88.9%) | 108 (87.8%) | 1.00 | 0.64 |
| | | TT | 14 (17.1%) | 11 (14.3%) | 0.81 (0.34–1.91) | | 14 (11.1%) | 15 (12.2%) | 1.21 (0.55–2.66) | |
| | Overdominant | CC/TT | 53 (64.6%) | 62 (80.5%) | 1.00 | 0.024 | 76 (60.3%) | 93 (75.6%) | 1.00 | 0.014 |
| | | CT | 29 (35.4%) | 15 (19.5%) | *0.44 (0.21–0.91)* | | 50 (39.7%) | 30 (24.4%) | *0.50 (0.29–0.88)* | |
| | | GA | 59 (38.6%) | 68 (52.3%) | *1.70 (1.05–2.74)* | 0.03 | 26 (46.4%) | 42 (5%) | 1.42 (0.69–2.89) | 0.34 |

Zhu et al. (2021), *PeerJ*, DOI 10.7717/peerj.12377

**Table 7** Distribution of *ULK1* and *PIK3CA* polymorphisms in population of different ethnicity and its association with the risk of MPA.

| SNP ID | Model | Geno type | Ethnicity = Han | | | *p* value | Ethnicity = Zhuang | | | *p* value |
|---|---|---|---|---|---|---|---|---|---|---|
| | | | Control | Case | OR (95% CI) | | Control | Case | OR (95% CI) | |
| *ULK1* rs4964879 | | AA | 73 (47.7%) | 43 (33.1%) | 1.00 | | 19 (33.9%) | 24 (32%) | 1.00 | |
| | Codominant | GA | 59 (38.6%) | 68 (52.3%) | 1.88 (1.12–3.17) | 0.054 | 26 (46.4%) | 42 (56%) | 1.24 (0.56–2.76) | 0.48 |
| | | GG | 21 (13.7%) | 19 (14.6%) | 1.48 (0.71–3.07) | | 11 (19.6%) | 9 (12%) | 0.67 (0.23–1.98) | |
| | Dominant | AA | 73 (47.7%) | 43 (33.1%) | 1.00 | | 19 (33.9%) | 24 (32%) | 1.00 | |
| | | GA/GG | 80 (52.3%) | 87 (66.9%) | *1.78 (1.09–2.90)* | *0.02* | 37 (66.1%) | 51 (68%) | 1.07 (0.50–2.28) | 0.86 |
| | Recessive | AA/GA | 132 (86.3%) | 111 (85.4%) | 1.00 | | 45 (80.4%) | 66 (88%) | 1.00 | |
| | | G/G | 21 (13.7%) | 19 (14.6%) | 1.06 (0.54–2.07) | 0.87 | 11 (19.6%) | 9 (12%) | 0.58 (0.22–1.55) | 0.28 |
| | Overdominant | AA/GG | 94 (61.4%) | 62 (47.7%) | 1.00 | | 30 (53.6%) | 33 (44%) | 1.00 | |
| | | GA | 59 (38.6%) | 68 (52.3%) | *1.70 (1.05–2.74)* | *0.03* | 26 (46.4%) | 42 (5%) | 1.42 (0.69–2.89) | 0.34 |
| *ULK1* rs9481 | | AA | 60 (39.2%) | 47 (36.1%) | 1.00 | | 21 (37.5%) | 25 (33.3%) | 1.00 | |
| | Codominant | AG | 72 (47.1%) | 54 (41.5%) | 0.94 (0.56–1.59) | 0.18 | 26 (46.4%) | 32 (42.7%) | 1.07 (0.49–2.37) | 0.62 |
| | | GG | 21 (13.7%) | 25 (22.3%) | 1.74 (0.88–3.45) | | 9 (16.1%) | 18 (24%) | 1.61 (0.59–4.41) | |
| | Dominant | AA | 60 (39.2%) | 47 (36.1%) | 1.00 | | 21 (37.5%) | 25 (33.3%) | 1.00 | |
| | | AG/GG | 93 (60.8%) | 83 (63.9%) | 1.12 (0.69–1.83) | 0.64 | 35 (62.5%) | 50 (66.7%) | 1.22 (0.58–2.54) | 0.6 |
| | Recessive | AA/AG | 132 (86.3%) | 104 (77%) | 1.00 | | 47 (83.9%) | 57 (76%) | 1.00 | |
| | | GG | 21 (13.7%) | 31 (23%) | 1.80 (0.96–3.35) | 0.064 | 9 (16.1%) | 18 (24%) | 1.55 (0.62–3.86) | 0.34 |
| | Overdominant | AA/GG | 81 (52.9%) | 79 (58.5%) | 1.00 | | 30 (53.6%) | 43 (57.3%) | 1.00 | |
| | | AG | 72 (47.1%) | 56 (41.5%) | 0.79 (0.49–1.27) | 0.33 | 26 (46.4%) | 32 (42.7%) | 0.91 (0.44–1.87) | 0.8 |
| *PIK3CA* rs1607237 | | CC | 76 (49.7%) | 86 (68.2%) | 1.00 | | 25 (45.5%) | 40 (56.3%) | 1.00 | |
| | Codominant | CT | 59 (38.6%) | 27 (21.4%) | *0.42 (0.24–0.73)* | *0.0068* | 20 (36.4%) | 18 (25.4%) | 0.59 (0.26–1.36) | 0.47 |
| | | TT | 18 (11.8%) | 13 (10.3%) | 0.68 (0.31–1.49) | | 10 (18.2%) | 13 (18.3%) | 0.81 (0.30–2.18) | |
| | Dominant | CC | 76 (49.7%) | 86 (68.2%) | 1.00 | | 25 (45.5%) | 40 (56.3%) | 1.00 | |
| | | CT/TT | 77 (50.3%) | 40 (31.8%) | *0.48 (0.29–0.78)* | *0.0031* | 30 (54.5%) | 31 (43.7%) | 0.67 (0.32–1.37) | 0.27 |
| | Recessive | CC/CT | 135 (88.2%) | 113 (89.7%) | 1.00 | | 45 (81.8%) | 58 (81.7%) | 1.00 | |
| | | TT | 18 (11.8%) | 13 (10.3%) | 0.91 (0.43–1.97) | 0.82 | 10 (18.2%) | 13 (18.3%) | 0.97 (0.38–2.51) | 0.96 |
| | Overdominant | CC/TT | 94 (61.4%) | 99 (78.6%) | 1.00 | | 35 (63.6%) | 53 (74.7%) | 1.00 | |
| | | CT | 59 (38.6%) | 27 (21.4%) | *0.44 (0.26–0.76)* | *0.0026* | 20 (36.4%) | 18 (25.4%) | 0.63 (0.28–1.38) | 0.24 |
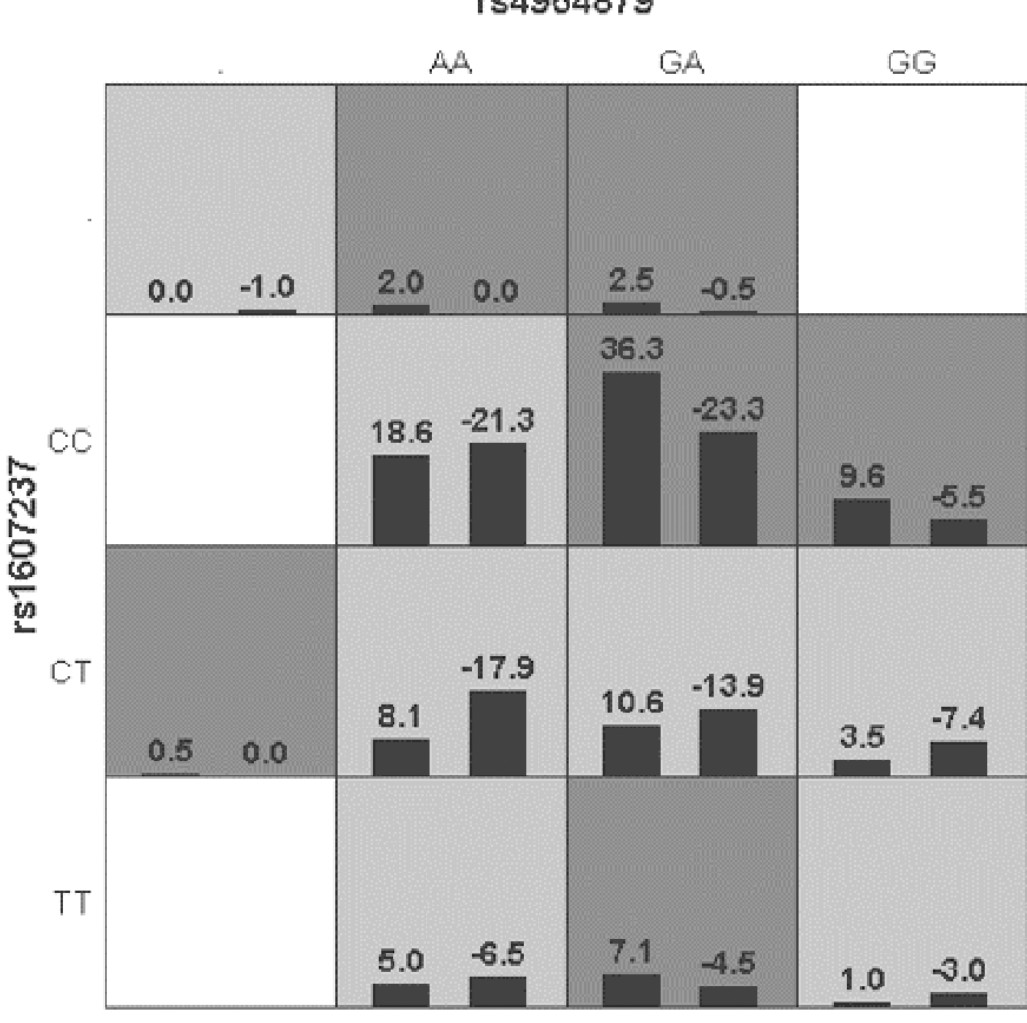

**Figure 2 Distribution of high-risk and low risk genotypes in the best two-locus model.** Dark gray and light gray boxed presented the high- and low- risk factor combinations, respectively. Left bars within each box represented case with positive score, right bars represented negative score. The higher the positive score, the greater the combination risk. GA in rs4964879 and CC in rs1607237 showed the most risk combinations.

In this study, six SNPs (rs10902469, rs12303764, rs4964879, rs7300908, rs7138581, and rs9481) in *ULK1* between healthy controls and MPA patients were evaluated. Our initial single SNP analysis detected a significant difference in the genotypic distribution (rs4964879, A >G) between the two groups. Subsequently, rs4964879 (codominant, dominant and overdominant model) and rs9481 (recessive model) of the *ULK1* gene were significantly associated with the risk of MPA. In addition, the association between *ULK1* gene polymorphisms and MPA risk was influenced by age, sex and ethnicity. Our findings indicated that rs4964879 and rs9481 variations (A>G) in the *ULK1* gene may be able to increase susceptibility to MPA.

*ULK1*, which is a serine/threonine protein kinase, plays a central role in initiating autophagy. It has been reported that the knockdown of *ULK1* in HEK293 cells is sufficient to inhibit the autophagy response (*Chan, Kir & Tooze, 2007*). Mouse embryonic fibroblasts derived from ULK1$^{-/-}$ and ULK2$^{-/-}$ mice blocked autophagy during amino acid starvation (*Cheong et al., 2011*). As expected by the role of ULK1 kinase in autophagy initiation, genetic variation in *ULK1* could result in autophagy disorder. David J. Horne et al. found that ULK1-deficient cells present decreased cytokine secretion and autophagy activity. The study was also the first to report that the rs12297124 minor allele of the *ULK1* gene contributes to an 80% reduction in latent tuberculosis infection risk in Asian participants (*Horne et al., 2016*). The *ULK1* SNPs rs4964879 and rs9481 reported in this study are located in intron and 3′UTR regions, respectively. Although introns are untranslated regions in mRNAs, mutations in introns may affect the binding of transcription factors and change the splicing modes or transcription of the *ULK1* gene, ultimately altering the sequence of amino acids (*Kawasaki et al., 2018*). The 3′UTR plays an important role in mRNA transport, stability and posttranscriptional regulation. Trans-acting factors or microRNAs bind to cis-acting elements in the 3′UTR of the target transcript and regulate protein synthesis by affecting transcription factors. Sequence variations in mRNA introns or 3′UTR regions in *ULK1* may cause abnormal expression of the gene (*Zhang et al., 2019*). Considering the role of ULK1 in the autophagy pathway, we speculate that *ULK1* (rs4964879 and rs9481) variations may lead to abnormal expression of ULK1 and then initiate the autophagy response, eventually increasing the susceptibility to MPA.

In the present study, another autophagy-related gene, *PIK3CA* (rs1607237, C>T), showed significant differences in the allele frequency and genotypic distribution between the patients with MPA and the healthy controls. A subject with at least one T allele has approximately half the risk for MPA compared with a subject with a CC genotype (TT+CT vs. CC: OR 0.56, 95% CI [0.37–0.83]). Increasing findings confirm that polymorphisms in the PI3K/AKT signaling pathway are related to the regulation of cell proliferation, survival and death. Similar to the results of our study, a case-control study conducted by Xing et al. found that the *PIK3CA* polymorphism is a defense factor against follicular thyroid cancer (*Xing et al., 2012*). *PIK3CA* rs1607237 is also significantly associated with a small decrease in breast cancer risk (*Stevens et al., 2011*). SNP rs1607237 is in the intron of *PIK3CA* gene. Although no published literature has reported the feature of *PIK3CA* rs1607237, given the location of this SNP, we speculate that SNPs may affect the transcription of the *PIK3CA* gene by interrupting the process of translation and splicing. The *PIK3CA* mutation increases the expression of the p110 $\alpha$ catalytic subunit of PI3K and then activates AKT through the PI3K/AKT signaling pathway. As mentioned above, the PI3K/AKT pathway positively regulates the mTOR/ULK1 pathway, and the activation of AKT may decrease the autophagy response. However, further research will be needed to provide strong evidence for this speculation.

This study has several limitations, which should be mentioned. First, the number of participants was relatively small because of the low incidence rate of MPA, especially for subpopulations after stratified analysis, which may provide insufficient evidence to provide definitive conclusions. Second, some follow-up information, such as the curative effect of

glucocorticoids and immunosuppressants, renal survival rate, relapse rate, and mortality, was lacking. Third, we did not perform studies to explore the molecular mechanisms to verify the association between gene polymorphisms reported in this study and MPA.

## CONCLUSIONS

The present study indicated that the polymorphisms observed in *ULK1* (rs4964879 and rs9481) and *PIK3CA* (rs1607237) were significantly associated with MPA risk in the Guangxi population. However, the molecular mechanisms are still unclear, and studies designed with larger samples and basic research are needed to confirm our conclusions and explore the mechanisms.

## ACKNOWLEDGEMENTS

We thank our colleagues in the Department of Nephrology, the Second Affiliated Hospital of Guangxi Medical University and the Experimental Center of Guangxi Medical University for their administrative and academic support. We also thank all participants enrolled in this study.

### Funding

This work was supported by funding from the Guangxi Natural Science Foundation Program (No. 2018GXNSFAA281122), the Development and Application Project of Medical and Health in Guangxi Zhuang Autonomous Region (No. S2017010), the NSFC cultivation project of The Second affiliated hospital of Guangxi Medical University (No. GJPY2018009). The funders had no role in study design, data collection and analysis, decision to publish, or preparation of the manuscript.

### Grant Disclosures

The following grant information was disclosed by the authors:
Guangxi Natural Science Foundation Program: 2018GXNSFAA281122.
Development and Application Project of Medical and Health in Guangxi Zhuang Autonomous Region: S2017010.
NSFC cultivation project of The Second affiliated hospital of Guangxi Medical University: GJPY2018009.

### Competing Interests

The authors declare there are no competing interests.

### Author Contributions

- Yan Zhu conceived and designed the experiments, performed the experiments, prepared figures and/or tables, authored or reviewed drafts of the paper, and approved the final draft.

- Jinlan Rao performed the experiments, analyzed the data, authored or reviewed drafts of the paper, and approved the final draft.
- Jingsi Wei performed the experiments, analyzed the data, prepared figures and/or tables, and approved the final draft.
- Liu Liu, Shanshan Huang and Jingjing Lan analyzed the data, prepared figures and/or tables, and approved the final draft.
- Chao Xue, Wei Li conceived and designed the experiments, authored or reviewed drafts of the paper, and approved the final draft.

## Human Ethics

The following information was supplied relating to ethical approvals (i.e., approving body and any reference numbers):

The study was approved by the Ethics Committee of the Second Affiliated Hospital of Guangxi Medical University (NO. 2018 KY-0100)

## Data Availability

The raw measurements are available in the Supplementary Files.

## Supplemental Information

Supplemental information for this article can be found online at http://dx.doi.org/10.7717/peerj.12377#supplemental-information.

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
