# Peer review of "Gene polymorphisms in ULK1 and PIK3CA are associated with the risk of microscopic polyangiitis in the Guangxi Zhuang Autonomous Region in China"

_PeerJ, doi:10.7717/peerj.12377_

## Round 0.1 · original submission · Major Revisions

Please address critiques of both reviewers and revise manuscript accordingly.

Reviewer 1 ·

Basic reporting

The manuscript “ Gene polymorphisms of ULK1 and PIK3CA are associated with risk of microscopic polyangiitis in the Guangxi Zhuang Autonomous Region in China” (#62693) addressed a rare type of vasculitis. The authors try to find a connection between SNPs in autophagy-related genes, such as ULK1 and PI3KCA and the susceptibility to microscopic polyangiitis.
The study included 208 eligible patients with microscopic polyangiitis and 211 healthy controls.
The manuscript is well written, includes a sufficient introduction. However, in this part is not clearly presented the authors` hypothesis, why they have focused exactly on these genes in their research.
The structure of the article conforms to the format of the journal. Figures are relevant to the content of the article, appropriately labelled. Raw data are shared, but they are partial and incomplete. These are not all the data based on which the analyzes in the article are made.

Experimental design

The study was conducted in conformity with the ethical standards in the field and the authors presented in supplementary data. Methods are described in detail.
I think that "Clinical Characteristics of study population" from section "Results" have to be moved to the section "Materials and Methods".

Validity of the findings

All underlying data are presented appropriately. There are discrepancies in the presentation of the results in the text and in the tables. There are not any results about the functional activity of these SNPs. Table 5 is difficult to trace and should be divided into three tables - by age, gender and ethnicity.
Conclusions are general. The study is descriptive but does not give any answers to finding even speculative ones.

Reviewer 2 ·

Basic reporting

This manuscript utilizes single-nucleotide polymorphism (SNP) analysis to investigate the associations between the autophagy-related genes ULK1, PIK3CA and the autoimmune disease-microscopic polyangiitis (MPA) in patients from the Guangxi Zhuang Autonomous Region in China. Through series of data analyses, they revealed two risk loci in ULK1 and one protective locus in PIK3CA for MPA. This idea behind the manuscript is meaningful for both basic and clinical research, however, there are certain gaps and misleading parts in result and discussion, which need to be further improved.

Major Concerns:
-line51-53, the result said “ULK1 haplotype (G-T-A-G-A) and PIK3CA haplotype (C-G) are associated with an increased risk of MPA, and PIK3CA haplotype (T-G) is associated with a reduced risk of MPA.” This conclusion mainly comes from the Table 4, the haplotype of ULK1 is G-T-A-C-G-A, not “G-T-A-G-A”. And it was associated with a reduced risk of MPA, not “an increased risk”. Please check the result carefully and draw the correct conclusion.

Minor Concerns:
The English language should be improved to ensure that an international audience can clearly understand your text. Some examples where the language could be improved:
-line44 & 124 : “high-through sequencing” should be “ high-throughput sequencing”.
-line47: “The single-SNP analysis detected significant differences in genotypic distribution in two SNPs between two groups: ULK1 rs4964879 (p = 0.019) and PIK3CA rs1607237 (p = 0.002).” please reorganize this sentence.
-line50: “reduced MPA risk” should be “ a reduced MPA risk”.
-line67: “;” should be “,”.
-line71: “A European study showed that PR3-ANCA is associated with HLA-DP that encodes SERPINA1 and PRTN3, where MPO-ANCA is associated with HLA-DQ (5).” Please refer to the original article, and reorganize this sentence, for example, should be the genes of anti–proteinase 3 ANCA encode SERPINA1 and PRTN3. “where” should be “while” in your sentence.
-line153: “consisted of 128 females and Han nationality population”, should be “consisted of 128 females and 155 Han nationality population”.
-line159: “the allele frequencies of gene PI3KCA rs1607237 (C>T) were significantly different between the MPA group and the control group (p = 0.013).” Based on the Table 2, the p value should be 0.011?
-line182: (Table 3) should be (Table 4).
-line200: “could increase MPA risk” should be “, which could increase MPA risk”.
-line235: “The knockdown of ULK1 in HEK293 cells was reported sufficient to inhibit autophagy response”. please reorganize this sentence.
-line278: “loci” should be “locus”

Experimental design

The authors mentioned the selection criteria for the SNPs in ULK1 and PIK3CA genes, the sites were located in the functional region. So, are there any solved structures for the proteins encoded by ULK1 and PIK3CA genes. If so, could the author further discuss the potential reasons why the SNPs in ULK1 and PIK3CA genes have different affects on MPA based on the protein structures. If not, are there literatures talking about the relationship or signaling pathways between ULK1 and PIK3CA-related proteins, please discuss it. The connections between ULK1 and PIK3CA genes, excepting they are both autophagy-related genes, could further explain the purpose of this study and experimental design.

Validity of the findings

In the conclusion, the authors mentioned basic research is needed to confirm their conclusion, which also should be one of the limitations in this study. Please discuss it in “Discussion Section”. For example, what types of basic research are needed to confirm your results, which could provide guidance for further studies.

---

## Round 0.2 · Minor Revisions

Although the reviewer was mostly satisfied by the revision, it was pointed out that your manuscript requires copyediting. Please note that copyediting is not provided as a standard publication service. Please ensure the language in this submission is clear and unambiguous, grammatically correct and conforms to professional standards of courtesy and expression.

Reviewer 2 ·

Basic reporting

The authors have sufficiently addressed my concerns.
However, the authors should carefully check the manuscript again to make sure professional English used throughout. For example, there are still wrongly written word "PI3KCA" in table 2 and 4.

Experimental design

The authors have sufficiently addressed my concerns.

Validity of the findings

The authors have sufficiently addressed my concerns.

---

## Round 0.3 · accepted · Accept

Thank you for addressing the remaining concerns of the reviewer and for careful editing of the manuscript. I am pleased to accept the revised manuscript now.